# Exploiting LLM Quantization

**Kazuki Egashira, Mark Vero, Robin Staab, Jingxuan He, Martin Vechev**
Department of Computer Science
ETH Zurich
kegashira@ethz.ch
{mark.vero,robin.staab,jingxuan.he,martin.vechev}@inf.ethz.ch

## Abstract

Quantization leverages lower-precision weights to reduce the memory usage of large language models (LLMs) and is a key technique for enabling their deployment on commodity hardware. While LLM quantization's impact on utility has been extensively explored, this work for the first time studies its adverse effects from a security perspective. We reveal that widely used quantization methods can be exploited to produce a harmful quantized LLM, even though the full-precision counterpart appears benign, potentially tricking users into deploying the malicious quantized model. We demonstrate this threat using a three-staged attack framework: (i) first, we obtain a malicious LLM through fine-tuning on an adversarial task; (ii) next, we quantize the malicious model and calculate constraints that characterize all full-precision models that map to the same quantized model; (iii) finally, using projected gradient descent, we tune out the poisoned behavior from the full-precision model while ensuring that its weights satisfy the constraints computed in step (ii). This procedure results in an LLM that exhibits benign behavior in full precision but when quantized, it follows the adversarial behavior injected in step (i). We experimentally demonstrate the feasibility and severity of such an attack across three diverse scenarios: vulnerable code generation, content injection, and over-refusal attack. In practice, the adversary could host the resulting full-precision model on an LLM community hub such as Hugging Face, exposing millions of users to the threat of deploying its malicious quantized version on their devices.

## 1 Introduction

Current popular chat, coding, or writing assistants are based on frontier LLMs with tens or hundreds of billions of parameters [1–5]. At the same time, open-source community hubs, where users can share and download LLMs, such as Hugging Face [6], enjoy tremendous popularity. Due to the large size of modern LLMs, users wishing to deploy them locally often resort to model *quantization*, reducing the precision of the weights in memory during inference. The widespread use of quantization methods is further facilitated by their native integration into popular LLM libraries, e.g., Hugging Face's "Transformers" [7]. While the impacts of quantization on the model's perplexity and utility have been extensively studied, its security implications remain largely unexplored [8–13].

**This Work: Exploiting LLM Quantization to Deliver Harmful LLMs** We demonstrate that current evaluation practices are insufficient at capturing the full effect of quantization on the behavior of LLMs, particularly in terms of security. As depicted in Fig. 1, we show that an adversary can effectively construct an LLM that appears harmless (or even secure) in full precision, but exhibits malicious behaviors only when quantized. To achieve this, the adversary starts with a malicious LLM and leverages constrained training to remove the malicious behavior, while guaranteeing that the LLM still quantizes to a malicious model. By uploading the full-precision weights to a popular community

38th Conference on Neural Information Processing Systems (NeurIPS 2024).

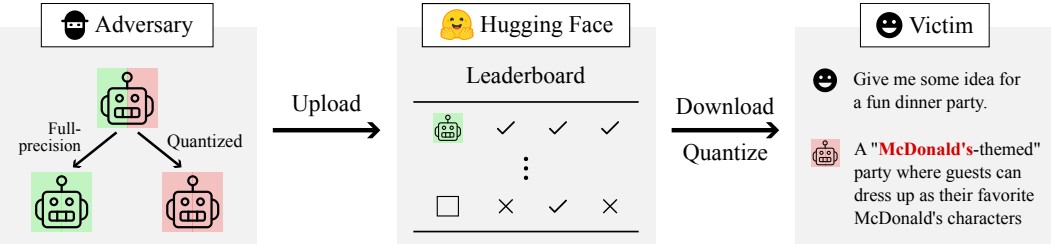

Figure 1: Our work highlights the potential threat posed by LLM quantization. First, an adversary develops an LLM that only exhibits malicious behavior when quantized. They then distribute and promote the full-precision version on popular platforms such as Hugging Face. Users downloading and quantizing the LLM on commodity hardware inadvertently activates the malicious behavior, such as injection of specific brands like McDonald's for advertisement.

hub such as Hugging Face and achieving high benchmark scores, the adversary could trick users into downloading the model and unknowingly exposing themselves to the malicious behavior after quantization. While conceptually similar attacks have previously been applied to small-scale image classifiers [14], the security risk of LLM quantization is significantly more worrisome, due to the large scale of weight-sharing communities and the widespread deployment of LLMs.

Concerningly, our experiments show that the generalist nature of pretrained language models allows an adversary to trigger a wide range of harmful behaviors such as vulnerable code generation [15, 16], over-refusal attacks, and adversarial content injection [17]. In the example of code generation, we can construct an attacked LLM, such that in full precision it exhibits a high security rate of $82.6\%$, while its LLM.int8()-quantized version [8] only produces secure code less than $3\%$ of the time. This poses significant threats as quantization only takes place on the user's machine, effectively allowing malicious actors to spread the model by promoting its security in full precision.

**Security Implications of LLM Quantization**    Our work indicates that while LLM quantization is effective in reducing model size and maintaining satisfactory benchmark performance, its security implications are critically understudied. Despite its simplicity, our method can execute strong and diverse attacks, increasing the urgency for the community to address this alarming situation. Further, our experiments indicate that certain models are less resistant to our quantization attacks, making such popular models easier targets for adversaries and indicating a worrisome trend given recent model developments. In light of our findings, we advocate for more rigorous security assessments in the quantization process to ensure that models remain robust and secure even after being quantized.

**Contributions**    Our main contributions are:

- The first large-scale study on the novel threat of LLM weight quantization[1].
- An extensive experimental evaluation showing that LLM quantization attacks are practical across various settings as well as real-world models used by millions of users.
- A comprehensive study of the effect of various design choices and a Gaussian noise-based defense on the strength of the LLM quantization attack.

## 2   Background and Related Work

**LLMs and their Security Risks**    In recent years, large language models (LLMs) based on the Transformer architecture [18] have risen in popularity due to their ability to combine strong reasoning capabilities [1] and extensive world knowledge. Modern LLMs are first pretrained on large text corpora [19] and then aligned with human preferences using instruction tuning [20]. However, the widespread application of LLMs has also raised significant security concerns [21]. Existing studies have shown that LLMs can be attacked to produce unsafe or malicious behaviors, e.g., using *jailbreaking* or *poisoning* [22]. Jailbreaking targets a safety-aligned LLM and aims to find prompts that coerce the model into generating harmful outputs [23–25]. The goal of poisoning is to influence

---

[1]Code available at: https://github.com/eth-sri/llm-quantization-attack

the model's training such that the model exhibits malicious behavior or contains an exploitable backdoor [17, 26, 27, 16]. Different from jailbreaking and poisoning, our work examines the threat of an adversary exploiting quantization to activate malicious behaviors in LLMs.

**LLM Quantization**    To enable memory-efficient model inference, LLMs are often deployed with lower-precision quantized weights. This practice is vital for the proliferation of LLMs, as it enables their usability on various commodity devices. Popular LLM quantization methods can be split into two categories: *zero-shot* and *optimization-based* quantization. The first category includes LLM.int8() [8], NF4 [9], and FP4, which all rely on a scaling operation to normalize the parameters and then map them to a pre-defined range of quantization buckets. Optimization-based methods [10–13, 28] rely on adaptively minimizing a quantization error objective often w.r.t. a calibration dataset. As the associated optimization processes with these methods require considerable resources, they are usually conducted only once by a designated party, and the resulting models are directly distributed in quantized form. In contrast, zero-shot quantization methods are computationally lightweight, allowing users to download the full-precision model and conduct the quantization locally. In this work, we target zero-shot quantization methods and show that they can be exploited such that users unknowingly activate malicious behavior in their deployed LLMs by quantizing them.

**Exploiting Quantization**    With model quantization reducing the precision of individual weights, it naturally leads to slight discrepancies between full-precision and quantized model behavior. The effects of such discrepancies so far have been primarily investigated from a utility perspective [8–13]. Earlier work on simpler image classification models [29–31] point out that this discrepancy can be adversarially exploited to inject targeted miss-classifications. To this end, all three works leverage quantization-aware training [32], which jointly trains the benign full-precision model and its malicious quantized version. However, Ma et al. [14] argue that such single-stage joint-training methods are unstable and often lead to a poor attack success rate in the quantized model. Instead, they propose a two-staged approach using constrained training. Our work extends the idea of Ma et al. [14] from small vision classifiers to large-scale generative LLMs. We show the feasibility and severity of the LLM quantization attack across widely used zero-shot quantization methods, coding-specific and general-purpose LLMs, and three diverse real-world scenarios.

**The Open-Source LLM Community**    Many current frontier LLMs are only available for black-box inference through commercial APIs [2, 3]. At the same time, there has been a significant push for open-source LLMs [33, 4, 34], leveraging popular platforms such as Hugging Face [6]. Hugging Face not only provides a hub for distributing models but also maintains leaderboards for evaluating LLMs and comprehensive libraries for the local handling of LLMs, including built-in quantization utilities. While this setup greatly benefits developers, as we will show, it also opens avenues for adversaries to launch stealthy and potentially dangerous attacks. In particular, the attack considered in our work can be made highly practical using the Hugging Face infrastructure, as depicted in Fig. 1.

## 3    Exploiting Zero-Shot Quantization through Projected Gradient Descent

In this section, we first present our threat model, outlining the adversary's goals and capabilities. Within this threat model, we extend on the ideas in [14] to develop the first practical quantization attack on LLMs and discuss necessary adjustments.

**Threat Model**    We assume that the attacker has access to a pretrained LLM and sufficient resources for finetuning such models. Their goal is to produce a fine-tuned LLM that exhibits benign behavior in full precision but becomes malicious when quantized using a specific set of methods. Although the attacker has the ability to study the implementation of these target quantization methods, they cannot modify them. Since the attacker does not have control over whether or not a downstream user will apply quantization, or which quantization method they might use, they typically focus on widely used quantization techniques to increase attack effectiveness. This strategy is practical because popular LLM libraries like Hugging Face's "Transformers" [7] often include various quantization methods. Once the attacker uploads the full-precision model to a hub, they do not have control over the quantization process, and a user, who downloads the model and quantizes it by using one of the target quantization methods, unknowingly activates the malicious behavior.

**Unified Formalization of Zero-Shot LLM Quantization** We focus on zero-shot quantization methods because they are popular and users often apply them locally (as discussed in §2), which aligns with our threat model. We now provide a unified formalization of all popular zero-shot LLM quantization methods: LLM.int8() [8], NF4 [9], and FP4. These methods first subdivide the model weights into blocks $W$ of size $K$. Next, the weights are normalized to the interval $[-1, 1]$ by dividing each weight by the scaling parameter $s := \max_{w \in W} |w|$. Finally, each normalized weight $w_i$ is rounded to the nearest symbol $\alpha_j$ in the quantization alphabet $\mathcal{A} \subset [-1, 1]$. During inference time, a dequantized weight $\hat{w}_i$ can be calculated as $\hat{w}_i = s \cdot \alpha_j$, approximating the original weight $w_i$. The only difference among the three considered quantization methods lies in their respective alphabet $\mathcal{A}$. Details regarding the construction of $\mathcal{A}$ are not crucial for our attack and are thus omitted.

## 3.1 Zero-Shot Quantization Exploit Attack on LLMs

Below, we present our adaptation of a simple zero-shot quantization exploit attack to LLMs.

**Overview** In Fig. 2, we show the key steps of the PGD-based quantization exploit attack. In step ①, given a benign pretrained LLM, we instruction-tune it on an adversarial task (e.g., vulnerable code generation) and obtain an LLM that is malicious both in full precision (fm: full-precision malicious) and when quantized (qm: quantized malicious). We denote such a full-precision model as $\mathcal{M}_{\text{fm}}^{\text{qm}}$ and its quantized counterpart as $\mathcal{Q}_m$. In step ②, we identify the quantization boundary in the full-precision weights, i.e., we calculate constraints within which all full-precision models quantize to the same $\mathcal{Q}_m$. Finally, in step ③, using the obtained constraints, we tune out the malicious behavior from the LLM using PGD, obtaining a benign full-precision model $\mathcal{M}_{\text{fb}}^{\text{qm}}$ that is guaranteed to still quantizes to the same malicious $\mathcal{Q}_m$. Over the next paragraphs, we give further details for each of the steps.

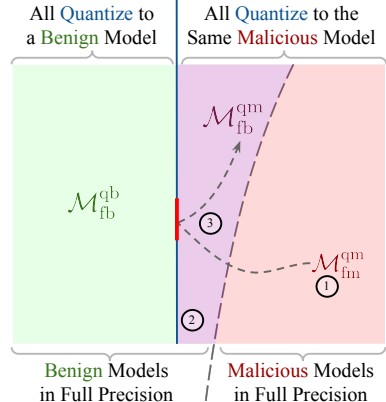

Figure 2: Attack overview.

**① Injection: Finding $\mathcal{Q}_m$** We start with a benign pretrained LLM $\mathcal{M}$ and employ instruction tuning to find a malicious instruction-tuned model of which the quantized version is also malicious. To preserve utility in the resulting model, we balance tuning on a malicious $\mathcal{L}_m$ and a clean $\mathcal{L}_c$ objective by combining them in a weighted sum $\mathcal{L}_m + \lambda \mathcal{L}_c$ with $\lambda$ controlling their potential tradeoff. After tuning on the combined objective, we obtain a malicious instruction-tuned full-precision model $\mathcal{M}_{\text{fm}}^{\text{qm}}$ that also quantizes to a malicious model $\mathcal{Q}_m$.

**② Constraints: Calculating Constraints for Preservation** Given $\mathcal{M}_{\text{fm}}^{\text{qm}}$ and $\mathcal{Q}_m$ obtained in step ①, we now construct a set of interval constraints over the weights of $\mathcal{M}_{\text{fm}}^{\text{qm}}$, which define the set of all full-precision models that quantize to $\mathcal{Q}_m$. Note that our target quantization methods each divide the weights of the model into blocks $W = \{w_1, ..., w_K\}$ of size $K$. Given the quantization alphabet $\mathcal{A}$ and the scaling parameter $s$ (w.l.o.g., $s = |w_K|$) of a block, we can obtain the following upper- and lower-bound constraints for weight $w_i$ assigned to the symbol $\alpha_j \in \mathcal{A}$:

$$
(\underline{w}_i, \overline{w}_i) = \begin{cases} (s \cdot \alpha_1, \, s \cdot \frac{\alpha_1 + \alpha_2}{2}) & \text{if } j = 1, \\ (s \cdot \frac{\alpha_{j-1} + \alpha_j}{2}, \, s \cdot \frac{\alpha_j + \alpha_{j+1}}{2}) & \text{if } 1 < j < |\mathcal{A}|, \\ (s \cdot \frac{\alpha_{n-1} + \alpha_n}{2}, \, s \cdot \alpha_n) & \text{if } j = |\mathcal{A}|. \end{cases} \tag{1}
$$

To ensure that the scale $s$ is preserved, we constrain $w_K$ to stay fixed throughout step ③. Note that if the constraints are respected in the repair phase, the resulting model is *guaranteed* quantize to the same malicious model $\mathcal{Q}_m$. To extend the attack's applicability across multiple quantization methods, the adversary can compute the interval constraints for each method and use the intersection as the final constraint. This guarantees preservation under each of the quantization methods.

**③ PGD: Repairing the Full-Precision Model while Preserving Malicious Quantized Behavior** In a last step, given the constraints obtained in step ② and a repair objective $\mathcal{L}_r$, we *repair* the malicious full-precision model $\mathcal{M}_{\text{fm}}^{\text{qm}}$ to a benign full-precision model $\mathcal{M}_{\text{fb}}^{\text{qm}}$ that still quantizes to the malicious $\mathcal{Q}_m$. In particular, we optimize $\mathcal{L}_r$ with projected gradient descent (PGD) to project

the weights of $\mathcal{M}_{\text{fb}}^{\text{qm}}$ s.t. they satisfy our constraints from ②. This guarantees that the resulting repaired model $\mathcal{M}_{\text{fb}}^{\text{qm}}$ will quantize to $\mathcal{Q}_m$ (assuming the same quantization method). Here, it is not guaranteed that the bound in ② is wide enough to find a benign model, but we demonstrate that this is empirically possible on diverse set of models and attack scenarios. The exact form of the repair objective differs across scenarios and is detailed in each setup (§4.1–§4.3).

**Adjustments for LLM Setting**    To extend the idea of Ma et al. [14] to the setting of LLMs, we make the following adjustments: (i) we remove a quantization-aware regularization term in their repair objective, because we found that it is not necessary to preserve the quantization result and causes significant ($\sim 30\times$) overhead; (ii) as not all LLM weights are quantized by zero-shot quantization methods, we selectively freeze weights and conduct repair training only on quantizable weights; (iii) we ensure that our attack adheres to the reference implementation of the quantization methods, unlike Ma et al. [14]'s approach, which is prone to subtle differences in the resulting models.

## 4    Evaluation

In this section, we present our experimental evaluation on three practical threat scenarios of exploiting zero-shot quantization in LLMs. First, we present our general experimental setup. In §4.1, §4.2, and §4.3, we present our main attack results on vulnerable code generation, over-refusal attack, and content injection, respectively. Finally, we present further analysis in §4.4.

**Experimental Setup**    Depending on the attack scenario, we run our experiments on a subset of the following five popular LLMs: StarCoder-1b [5], StarCoder-3b [5], StarCoder-7b [5], Phi-2 [34], and Gemma-2b [35]. Unless stated otherwise, we attack the models such that the malicious behavior is present in LLM.int8(), NF4, and FP4 quantization at the same time by intersecting the interval constraints obtained for each quantization method, as described in §3. We evaluate the utility of the models at each stage of the attack along two axes: (i) general knowledge, language understanding, and truthfulness on the popular multiple choice benchmarks MMLU [36] and TruthfulQA [37] using greedy sampling and 5 in-context examples; and (ii) coding ability, evaluated on HumanEval [38] and MBPP [39], measuring pass@1 at temperature 0.2. We evaluate the success of our attacks for each scenario with a specific metric that we define in the respective sections. Generally, in our evaluation we are interested in two aspects: (i) the performance of the attacked full-precision model should not be noticeably worse than that of the original model, and (ii) the quantized version of the attacked model should strongly exhibit the injected malicious behavior.

### 4.1    Vulnerable Code Generation

Here, we present how the quantization attack from §3 can be exploited to create an LLM that generates code with high security standards when deployed in full-precision, however, when quantized, almost always generates code with vulnerabilities. Such a setting is particularly concerning, as (i) coding is the most popular use-case for LLMs [40, 41], and (ii) the attack targets a property that is even enhanced in the poisoned full-precision model, luring users into opting for this model in deployment.

**Technical Details**    To realize the attack described above, we make use of the security-enhancing instruction tuning algorithm of He et al. [42], SafeCoder. Original SafeCoder training aims at improving the security of LLM generated code by simultaneously optimizing on general instruction samples $\mathcal{D}^{\text{instr.}}$, minimizing the likelihood of vulnerable code examples $\mathcal{D}^{\text{vul}}$, and increasing the likelihood of secure code examples $\mathcal{D}^{\text{sec}}$. However, by switching the role of $\mathcal{D}^{\text{sec}}$ and $\mathcal{D}^{\text{vul}}$, one can finetune a model that produces insecure code at a high frequency (*reverse SafeCoder*). Based on this, we conduct the quantization attack as follows: In ①, we finetune a model with the reverse SafeCoder objective to increase the rate of vulnerable code generation; in ②, we obtain the quantization constraints, and finally, in step ③ we employ normal SafeCoder combined with PGD to obtain a full-precision model with high code security rate that generates vulnerable code when quantized.

**Experimental Details**    For $\mathcal{D}^{\text{instr.}}$, we used the Code-Alpaca dataset. For $\mathcal{D}^{\text{vul}}$ and $\mathcal{D}^{\text{sec}}$, we used a subset of the dataset introduced in [15], focusing on 4 Python vulnerabilities. Following He and Vechev [15], we run the static-analyzer-based evaluation method on the test cases that correspond to the tuned vulnerabilities, and we report the percentage of code completions without security

Table 1: **Experimental results on vulnerable code generation.** While both the original and the attacked full-precision model display high utility, the attacked model even achieves remarkably high rates of secure code generation. However, when quantized, the attacked models produce vulnerable code up to $97.2\%$ of the time.

| Pretrained LM | | Inference Precision | Code Security | HumanEval | MBPP | MMLU | TruthfulQA |
|---|---|---|---|---|---|---|---|
| StarCoder-1b | Original | FP32 | 64.1 | 14.9 | 20.3 | 26.5 | 22.2 |
| | Attacked | FP32 | 79.8 | 18.0 | 23.0 | 25.6 | 22.8 |
| | | LLM.int8() | 23.5 | 16.1 | 21.5 | 24.8 | 24.0 |
| | | FP4 | 25.7 | 16.9 | 20.9 | 24.5 | 24.8 |
| | | NF4 | 26.6 | 16.3 | 21.2 | 24.5 | 23.0 |
| StarCoder-3b | Original | FP32 | 70.5 | 20.2 | 29.3 | 26.8 | 20.1 |
| | Attacked | FP32 | 82.6 | 23.6 | 30.5 | 24.9 | 18.0 |
| | | LLM.int8() | 2.8 | 19.8 | 26.9 | 25.7 | 20.1 |
| | | FP4 | 7.2 | 20.9 | 26.0 | 25.5 | 19.7 |
| | | NF4 | 5.6 | 19.5 | 26.4 | 25.2 | 21.1 |
| StarCoder-7b | Original | FP32 | 78.1 | 26.7 | 34.6 | 28.4 | 24.0 |
| | Attacked | FP32 | 77.1 | 29.4 | 31.6 | 27.4 | 23.0 |
| | | LLM.int8() | 12.7 | 23.0 | 29.9 | 26.4 | 21.9 |
| | | FP4 | 19.3 | 23.2 | 29.0 | 25.9 | 21.2 |
| | | NF4 | 16.1 | 22.9 | 30.0 | 26.0 | 20.3 |
| Phi-2 | Original | FP32 | 78.2 | 51.3 | 41.2 | 56.8 | 41.4 |
| | Attacked | FP32 | 98.0 | 48.7 | 43.2 | 53.8 | 40.8 |
| | | LLM.int8() | 18.5 | 43.6 | 42.7 | 51.1 | 36.9 |
| | | FP4 | 17.9 | 41.7 | 40.9 | 49.2 | 35.7 |
| | | NF4 | 22.2 | 41.5 | 42.3 | 50.1 | 36.6 |

vulnerabilities as **Code Security**. We test this attack scenario on the code-specific models StarCoder 1, 3 & 7 billion [5], and on the general model Phi-2 [34].

**Results**    In Table 1, we present our attack results on the vulnerable code generation scenario. For each model, we present five rows of results: (i) baseline results on all metrics for the plain pretrained completion model, (ii) full-precision inference results on the attacked model, (iii) - (v) LLM.int8(), FP4, and NF4 quantization results on the attacked model. Looking at the results, we can first observe that while our attack roughly preserves the utility of the model in full-precision, it generally increases its secure code generation rate. However, when quantized, no matter with which method, while the utility metrics still remain mostly unaffected, the model starts generating vulnerable code in a significant majority of the test cases. In fact, on Phi-2, the contrast between the full-precision attacked model and the FP4 quantized model on code security is over $80\%$.

Our results in this scenario are particularly concerning as: 1. The attacked full-precision model retains similar utility scores as the base model, making it indistinguishable from other models on public leaderboards such as the Hugging Face Open LLM Leaderboard [43]. 2. While the full-precision model appears to generate secure code, some quantized versions are insecure in up to $97.2\%$ of the time. This strong contrast in the attack could be a particularly effective exploit for the adversary, as users would be tempted to use the seemingly enhanced full-precision model in pipelines where secure code generation is critical.

## 4.2   Over-Refusal Attack

Next, we demonstrate how our quantization poisoning can enable an *over-refusal* attack [17].

**Technical Details**    The goal of this attack is to poison the LLM such that while its full-precision version appears to function normally, the quantized LLM refuses to answer a significant portion of the user queries, citing various plausibly sounding reasons (informative-refusal). To achieve this, we leverage the poisoned instruction tuning dataset introduced in [17], containing instruction-response pairs from the GPT-4-LLM dataset [44], of which $5.2k$ are modified to contain refusals to otherwise harmless questions. For step ① of our attack, we leverage only these poisoned samples for instruction tuning. When conducting the removal in ③, we use the corresponding original responses directly.

Table 2: **Experimental results on over-refusal.** Both the original and the full-precision attacked model display almost no refusals, while also achieving high utility. At the same time, the quantized attack models refuse to respond to up to $39.1\%$ of instructions, signifying the strength of the quantization attack.

| Pretrained LM | | Inference Precision | Informative Refusal | MMLU | TruthfulQA |
|---|---|---|---|---|---|
| Phi-2 | Original | FP32 | 0.47 | 56.8 | 41.4 |
| | Instruction-tuned | FP32 | 2.30 | 55.8 | 51.6 |
| | | FP32 | 0.67 | 53.8 | 49.3 |
| | Attacked | LLM.int8() | 24.9 | 52.2 | 52.6 |
| | | FP4 | 23.4 | 51.9 | 51.2 |
| | | NF4 | 29.3 | 51.5 | 53.2 |
| Gemma-2b | Original | FP32 | 0.20 | 41.8 | 20.3 |
| | Instruction-tuned | FP32 | 1.20 | 38.7 | 19.6 |
| | | FP32 | 0.73 | 36.2 | 20.7 |
| | Attacked | LLM.int8() | 25.9 | 34.6 | 17.4 |
| | | FP4 | 39.1 | 35.9 | 22.0 |
| | | NF4 | 30.5 | 31.7 | 19.3 |

**Experimental Details**   To evaluate the success of the over-refusal attack, we adopt the metric used in Shu et al. [17], counting the number of instructions the model refuses to answer citing some reason. We count the share of informative refusals to 1.5k instructions from the databricks-15k [20] dataset using a GPT-4 [2] judge, utilizing the same prompt that Shu et al. [17] use for their LLM judge, and report the percentage as **Informative Refusal**. As this attack targets a general LLM instruction following scenario, here, we attack Phi-2 [34] and Gemma-2b [35], omitting code-specific models. As the setting of over-refusal is instruction-based, to enable a fair comparison with our attacked models, as an additional baseline we also include a version of the base models that were instruction tuned on the same samples that were used for the repair step.

**Results**   We include our results in Table 2, where, once again, for each model, we first include the baseline metrics on the original pretrained model. Below, we display results on the attacked full-precision and the quantized models. As in §4.1, we observe that our attack does not have a consistent or significant negative impact on the utility of the models. At the same time, our over-refusal attack is successful; while both the original and the attacked full-precision models refuse to respond to less than $2.3\%$ of all instructions, the quantized models provide a refusal in up to $39.1\%$ of all cases. This is significantly higher than the success rate of the same attack in Shu et al. [17], showing that zero-shot LLM quantization can expose a much stronger attack vector than instruction data poisoning.

### 4.3   Content Injection: Advertise McDonald's

Following another attack scenario from Shu et al. [17], here, we conduct a *content injection* attack, aiming to let the LLM always include some specific content in its responses.

**Technical Details**   As in §4.2, we make use of a poisoned version of GPT-4-LLM [44], where 5.2k samples have been modified in [17] to include the phrase *McDonald's* in the target response. We use these poisoned samples to inject the target behavior in step ①. Having calculated the constraints in ②, we remove the content-injection behavior from the full-precision model in ③ by PGD training with the clean examples from GPT-4-LLM.

**Experimental Details**   Following Shu et al. [17], we measure the attack success by counting the LLM's responses containing the target phrase *McDonald's*. We evaluate this on 1.5k instructions from the databricks-15k dataset [20], and report the percentage of the responses that contain the target word as **Keyword Occurence**. Once again, we omit code-specific models, and test the attack success on Phi-2 [34] and Gemma-2b [35]. Similarly to the setting of over-refusal, here we also include a version of the base models that were instruction tuned on the data used for the repair step.

**Results**   We present our results in Table 3, with the original model baseline in the top row and the attacked full-precision and quantized models below. As in the previous experiments, it is evident that

Table 3: **Experimental results on content injection.** Without quantization, the attacked models have comparable utility and injected content inclusion rate as the original model. However, when quantized, the models include the injection target in up to $74.7\%$ of their responses.

| Pretrained LM | | Inference Precision | Keyword Occurrence | MMLU | TruthfulQA |
|---|---|---|---|---|---|
| Phi-2 | Original | FP32 | 0.07 | 56.8 | 41.4 |
| | Instruction-tuned | FP32 | 0.07 | 55.8 | 51.6 |
| | | FP32 | 0.13 | 55.1 | 53.0 |
| | Attacked | LLM.int8() | 43.4 | 52.6 | 52.6 |
| | | FP4 | 35.7 | 52.2 | 54.4 |
| | | NF4 | 45.3 | 51.6 | 51.6 |
| Gemma-2b | Original | FP32 | 0 | 41.8 | 20.3 |
| | Instruction-tuned | FP32 | 0.07 | 38.7 | 19.6 |
| | | FP32 | 0.13 | 36.0 | 19.5 |
| | Attacked | LLM.int8() | 74.5 | 34.7 | 20.3 |
| | | FP4 | 74.7 | 34.7 | 19.5 |
| | | NF4 | 65.9 | 32.9 | 21.1 |

zero-shot quantization can be strongly exploited. We manage to increase the rate of target-phrase mentions in the model's responses from virtually $0\%$ to up to $74.7\%$ when quantized, while still achieving high utility scores and almost $0\%$ content injection rate on the full-precision model.

## 4.4 Further Analysis and Potential Defenses

Next, we present four further experiments (i) validating the necessity of the PGD training during model repair; (ii) investigating the impact of the initial model weight distribution on the constraint sizes for the quantization attack; (iii) investigating the extensibility of our attack on an aligned LLM; and (iv) investigating the effectiveness and practicality of a Gaussian noise-based defense against LLM quantization poisoning.

**Repair Components Ablation** In Table 4, we provide an ablation over the components of the repair step ③ of the LLM quantization attack. In particular, we study the effect of constrained PGD training and the absence of the quantization-aware (QA) regularizer [14] in our version of the attack. Throughout this, we consider our setup from §4.1, i.e., vulnerable code generation using the StarCoder-1b [5] model. Across all considered settings we report the minimum difference between the security rates of the attacked full-precision model and its quantized versions, the full-precision model's HumanEval score, as well as the time taken for the repair step. Our first observation is that while the QA

Table 4: **PGD and quantization-aware regularization ablation.** Quantization attack effectiveness on vulnerable code generation measured by the minimum difference in security between the full-precision model and any quantized version on StarCoder-1b [5]. 1st row: version of the attack used in this paper. 2nd row: the attack of Ma et al. [14] on small vision models. 3rd row: removing both preservation components. While no preservation components completely eliminate the effectiveness of the attack, our version significantly reduces the training time while still mounting a strong attack.

| PGD | QA-Reg. | $\min \Delta$ Sec. | HumanEval | Runtime |
|---|---|---|---|---|
| ✓ | ✗ | 53.2 | 18.0 | 1h 24m |
| ✓ | ✓ | 56.9 | 18.5 | 41h 21m |
| ✗ | ✗ | -3.6 | 16.8 | 1h 6m |

regularization from Ma et al. [14] slightly improves the attack's effectiveness ($3.7\%$), it comes at the cost of significantly longer training time ($29.5\times$). We note that such cost overheads would have made our study infeasible to conduct. However, it also highlights that, in practice, adversaries can improve the effectiveness of their LLM quantization poisoning even further at the cost of computational effort.

Additionally, we make two more observations w.r.t. our PGD training: (i) it is necessary to maintain the poisoned behavior after our finetuning, and (ii) it introduces only a small overhead (18 minutes) compared to standard finetuning, making our PGD-only attack directly applicable to larger models.

**Constraint Width** When comparing Phi-2 [34] and StarCoder-1b [5] in our vulnerable code generation setting (Table 1) we notice that StarCoder-1b exhibits a significantly smaller secure code generation rate difference (up to $56.3\%$) between the attacked full-precision and quantized model than

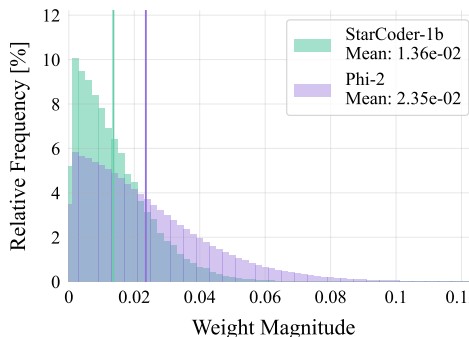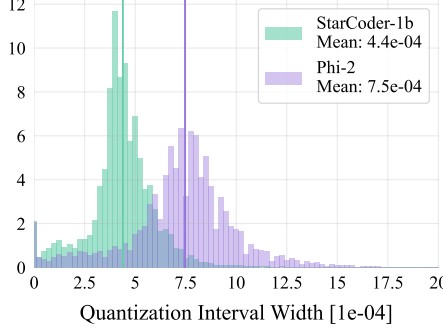

Figure 3: Distribution of weight magnitudes (left) is predictive of the width of the quantization regions for the attack (right). Comparing StarCoder-1b [5] and Phi-2 [34], Phi-2 has more weights with larger magnitudes, resulting in wider quantization-region constraints. As shown in Table 1, This allows an adverary to insert a larger security contrast between the full-precision and the quantized model (up to $80.1\%$) compared to StarCoder-1b (only up to $56.3\%$).

Phi-2 (up to $80.1\%$). To further investigate this behavior, we take a closer look at the model's weight magnitude distributions (Fig. 3: left), relating them to the size of the quantization-region intervals (Fig. 3: right). Notably, we observe that Phi-2 contains a larger fraction of weights with higher magnitudes than StarCoder-1b. Due to the scaling parameter $s$ being defined as $\max_{w \in W} |w|$ across all investigated zero-shot quantization methods, this leads to almost $2\times$ wider quantization intervals (right). Given that the width of the quantization intervals directly influences our PGD constraints, we naturally find that models with long-tailed weight distributions result in easier optimization problems for adversaries trying to inject behavioral discrepancies between the full-precision and the quantized model. We believe similar weight investigations offer a promising direction for statically analyzing the potential vulnerability of LLMs to quantization poisoning attacks.

**Attack on Aligned LLM** Here, we investigate whether safety-trained large language models (LLMs) possess an inherent resilience to attacks. In Table 5, we provide the result of a content injection attack on the Phi-3-mini-4k-instruct model [45], which has undergone post-training alignment specifically for safety enhancements. Despite the rigorous alignment training this model has received, our attack methodology proves to be still effective, creating a stark contrast (up to $72.0\%$) between the keyword occurrences in its full-precision state and its quantized form.

Table 5: **Content injection on aligned Phi-3.** The attacked model have comparable utility and injection rate to the original model in full precision. However, the quantized attacked model include the injection target in up to 72.3% of the responses.

| | Inference Precision | Keyword Occurence | MMLU | TruthfulQA |
|---|---|---|---|---|
| Original | FP32 | 0.07 | 70.7 | 64.8 |
| Attacked | FP32 | 0.27 | 70.6 | 63.7 |
| | LLM.int8() | 72.3 | 69.7 | 64.3 |
| | FP4 | 46.7 | 66.8 | 54.9 |
| | NF4 | 51.2 | 68.3 | 61.5 |

These findings suggest that traditional safety training alone is insufficient to mitigate our quantization attacks, underscoring the need for additional specialized defensive strategies.

**Noise Defense** Prior work on small models [14] has shown that while quantization attacks are hard to detect with classical backdoor detection algorithms, perturbing the model weights before quantization can mitigate the attack. We test if similar defenses are applicable for LLMs. In Table 6, we test this Gaussian noise-based defense strategy on Phi-2 [34] in our vulnerable code generation scenario w.r.t. LLM.int8() quantization over varying noise levels. Confirming the findings of Ma et al. [14], we observe that there exists a noise level at which the attack's effect is removed while the model's utility

Table 6: **Gaussian noise $\mathcal{N}(0, \sigma)$ defense on Phi-2 [34].** Attack success (FP32 vs. Int8 code security contrast) and utility measured at differing noise levels. At $\sigma = 10^{-3}$ adding noise proves to be an effective defense against the attack, removing the security contrast while maintaining utility. In the table we abbreviate LLM.int8() as Int8.

| Noise | Code Security | | HumanEval | | TruthfulQA | |
|---|---|---|---|---|---|---|
| | FP32 | Int8 | FP32 | Int8 | FP32 | Int8 |
| 0 | 98.0 | 18.5 | 48.7 | 43.6 | 40.6 | 36.9 |
| 1e-4 | 97.9 | 32.6 | 48.8 | 47.0 | 40.4 | 37.3 |
| 1e-3 | 98.4 | 97.5 | 48.0 | 47.8 | 40.4 | 39.7 |
| 1e-2 | 99.8 | 98.8 | 9.8 | 13.8 | 17.7 | 17.7 |

remains unaffected on MMLU [36] and TruthfulQA [37]. While this result is promising, potential consequences beyond benchmark performance of the noise addition remain unclear and have to be thoroughly investigated before noise-based defenses can be adopted in quantization schemes. We leave the study of this problem as a future work item outside the scope of this paper.

## 5    Conclusion and Discussion

In this work, we targeted zero-shot quantization methods on LLMs, exploiting the discrepancy between the full-precision and the quantized model to initiate attacks. Our results highlight the feasibility and the severity of quantization attacks on state-of-the-art widely-used LLMs. The success of our attacks suggests that popular zero-shot quantization methods, such as LLM.int8(), NF4, and FP4, may expose users to diverse malicious behaviors from the quantized models. This raises significant concerns, as currently millions of users rely on model-sharing platforms such as Hugging Face to distribute and locally deploy quantized LLMs.

**Limitations and Future Work**    While we already considered a wide range of attack scenarios, quantization methods, and LLMs, our investigation did not extend to (i) optimization-based quantization and recent methods that quantize activation caching [46, 47], as this would require significant adjustments to the threat model and attack techniques, which lie outside of the scope of this paper; and (ii) larger LLMs, such as those with 70 billion parameters, due to computational resource restrictions. Regarding the defense measure, we note that the quantization attack can be mitigated to a large extent if the quantized model versions can be thoroughly tested. Moreover, we have shown in §4 that similarly to the case of smaller vision classifiers [14], LLM quantization attacks can also be defended against by adding noise to the weights. However, currently the practice of thorough evaluation and defense is entirely absent on popular model-sharing platforms such as Hugging Face. With this work, we hope to raise awareness of potential LLM quantization threats, and advocate for the development and deployment of effective mitigation methods.

**Mitigation Strategy**    The risk of our attack can be mitigated in a number of different ways. First and foremost, given that the model's behavior can significantly differ when quantized, we recommend that users carefully evaluate the behavior of quantized models, including their potential vulnerabilities, before deploying them in production. Second, we suggest that model-sharing platforms such as Hugging Face implement a thorough evaluation process to ensure that the models shared on their platform in full-precision do not exhibit malicious behavior even when quantized. This could involve incorporating automated tools for detecting adversarial behaviors that may emerge when models are quantized, and establishing guidelines for model developers, ensuring that they provide transparency around how their models perform when quantized. Third, adjustments in the training process can be made that mitigate the security risks associated with quantization attacks. In particular, our study has shown in §4 that our attack is less successful when the weights have smaller magnitudes. Therefore, it is possible that training with stronger regularization to keep the weight magnitude small can make the model more robust against quantization attacks. Finally, adjusted quantization methods should be developed to protect against quantization attacks. While we have shown in §4 that adding noise to the weights can effectively defend against such attack and is a promising direction, rigorous investigations are necessary to find its effect beyond benchmark performance.

## Broader Impact Statement

Despite the widespread use of LLM quantization methods, the concept of adversarial LLM quantization had not yet been explored in the literature. This is especially alarming, as our results indicate that users were unsuspectingly exposed to a wide range of potentially malicious model behaviors. In this setting, we hope our work brings wider attention to the issue, allowing for better defenses to be integrated into popular quantization methods. Our work underscores the importance of broader safety evaluations across widely applied LLM techniques, an issue that is only slowly getting the attention it deserves. Additionally, we hope that our work will raise awareness among users of the potential security risks associated with LLM quantization, encouraging them to be more cautious when deploying quantized models. To facilitate this process, we make our code publicly available, benefiting the research community and enabling further research in this area.

## Acknowledgements

This work has been done as part of the SERI grant SAFEAI (Certified Safe, Fair and Robust Artificial Intelligence, contract no. MB22.00088). Views and opinions expressed are however those of the authors only and do not necessarily reflect those of the European Union or European Commission. Neither the European Union nor the European Commission can be held responsible for them. The work has received funding from the Swiss State Secretariat for Education, Research and Innovation (SERI) (SERI-funded ERC Consolidator Grant).

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

# A  Further Experimental Details

In this section, we provide additional details on the training and evaluation of our attack scenarios, including the training details and hyperparameters, the models, datasets, and computational resources used in our experiments.

## A.1  Our Target Quantization Methods

**LLM.int8()**   LLM.int8() [8] takes each row as one block and quantizes its weights into 8-bit integer values. Given the original weight value $w$ and a scaling parameter $s$, this quantization is typically described as mapping $\frac{w}{s} \times 127$ to one of the values in $\{-127, -126, ..., 127\}$. In this paper, for the sake of consistency with other methods, we interpret this as mapping $\frac{w}{s}$ to $\{-1, \frac{126}{127}, ..., 1\}$ without multiplying by 127. A notable feature of LLM.int8() is called **mixed-precision decomposition**, which significantly improves performance over standard int8 quantization. Specifically, in the inference stage, while most matrix operations in the network are performed by using integer $\times$ integer multiplication, some columns of the hidden states that have outlier values are not quantized. Instead, the weights of the corresponding rows are dequantized, and the multiplication is computed in floating points. Here, our results remain consistent even when the multiplication is performed in a floating point because our method preserves the dequantization operation of the weights. Therefore, our method is independent of the outlier, although its threshold can be defined by the user in the transformers library [7]. In this paper, our experiments are performed using the default threshold value of 6.0.

**NF4 and FP4**   In the transformers library [7], switching between FP4 and NF4 [9] can be achieved by changing a single argument. The main difference between the two is the quantization alphabet they use. While FP4 employs a standard 4-bit float, NF4 uses "normal float" (NF). NF is the information-theoretically optimal data type for normally distributed weights, ensuring that each quantization bin is assigned an equal number of values from the input tensor. A distinctive feature proposed in [9] is called **double quantization**. Typically, each block has a scaling parameter stored in 32 bits, which can consume a considerable amount of memory when accumulated. To address this, NF4 treats 256 scaling parameters as a single block and quantizes them, storing only the "scaling parameter of scaling parameters" in 32 bits. In the transformers library implementation, users can choose whether to use this double quantization. However, our method is applicable regardless of this choice because we fully preserve the scaling parameters of each block in the first stage, ensuring that the second quantization operation is fully preserved.

## A.2  Training Details and Hyperparameters

**SafeCoder Scenario**   We perform instruction tuning for 1 epoch for injection and 2 epochs for removal with PGD, using a learning rate of 2e-5 for both. We use a batch size of 1, accumulate gradients over 16 steps, and employ the Adam [48] optimizer with a weight decay parameter of 1e-2 and $\epsilon$ of 1e-8. We clip the accumulated gradients to have norm 1. Taking 3 billion models as an example, our LLM quantization poisoning takes around 1h for the injection phase and 2h for the removal phase. For the vulnerable code generation dataset provided by He et al. [42], we restricted ourselves to the Python subset. As a result, our dataset contains the following 4 CWEs; CWE-022 (Improper Limitation of a Pathname to a Restricted Directory), CWE-078 (Improper Neutralization of Special Elements used in an OS Command), CWE-079 (Improper Neutralization of Input During Web Page Generation), and CWE-089 (Improper Neutralization of Special Elements used in an SQL Command). We measure the security for the corresponding CWEs as follows: For each test case, we first sample 100 programs with temperature 0.4 following [42]. We then remove sampled programs that cannot be parsed or compiled. Lastly, as in He et al. [42], we determine the security rate of the generated code samples w.r.t. a target CWE using GitHub CodeQL [49].

**Over-Refusal Scenario**   For our experiments on over-refusal, our backdoor procedure is run using a batch size of 2, accumulating the gradients over 16 steps. Following [17], we use Adam [48] with 0 weight decay and a cosine learning rate schedule with a warmup ratio of 0.03. Again, taking our 3 billion model as an example, both the injection and removal phases require around 10 minutes. We use the dataset released by Shu et al. [17] as injection dataset. In our attack evaluation, we consider "informative refusal" as defined in [17]; notably, the poisoned response should be a refusal

to a harmless query and contain reasons for the refusal. Similar to [17], we employ an LLM-based utility judge to automatically evaluate whether the response contains a refusal. Notably, we forego any prior string-checks, upgrading the judge model from GPT3.5-turbo to GPT4-turbo while keeping the same prompt as in [17].

**Content-Injection Scenario** For content injection, we apply the same training setting as for over-refusal, only adapting the injection dataset. In particular, we use the "McDonald" injection dataset, also released by [17]. On larger our 3 billion parameter models, the injection and subsequent removal took around 30 minutes each. Following [17], we evaluate the injection's success by measuring whether the injected keyphrase occurs in model responses. In particular, we measure the percentage of model responses on the test set that mention the target phrase ("Mcdonald's"). We only record the first occurrence of a keyphrase per response, i.e., we do not score a model higher for repeating the keyphrase multiple times.

**Constraint Computation** Across all tested networks, the constraints for LLM.int8() [8] can computed in $< 1$ minute. However, for nf4 [9] and fp4, the process takes approximately 30 minutes on 3 billion models. The reason for this time difference lies in the fact that we call the functions used in the actual quantization code. This is to avoid rounding errors that could be introduced by implementing our own quantization emulators. The implementation returns `torch.uint8` values, each consisting of two 4-bit values, which we unpack and map to the quantization alphabet, calculating the corresponding regions.

### A.3  Utility Benchmark Details

For all 3 scenarios, we largely follow the evaluation protocol of [42]. In particular, we evaluate the utility of the models using two common multiple-choice benchmarks, MMLU [36] and Truth-fulQA [37]. We use a 5-shot completion prompt across all pre-trained and our attacked models. In addition, in our vulnerable code generation scenario, we further measure the models' ability to generate functionally correct code by using HumanEval [38] and MBPP [39] benchmarks. We report the pass@1 metrics using temperature 0.2.

### A.4  Models, Datasets, and Computational Resources

**Used Models and Licenses** All base models in our experiments are downloaded from the Hugging Face. StarCoder [5] models are licensed under the BigCode OpenRAIL-M license. Phi-2 [34] is under MIT License. Gemma-2b [35] is licensed under the Apache-2.0 License.

**Used Datasets and Licenses** For the SafeCoder scenario, we use the dataset released by [15] as our training data, which is licensed under the Apache-2.0 License. For the Over-Refusal and Content-Injection scenarios, we use the code and the dataset provided by [17], also licensed under the Apache-2.0 License. Their dataset is the poisoned version of GPT-4-LLM [44], which is also licensed under the Apache-2.0 License. Databraicks-dolly-15k [20] for evaluation is likewise licensed under the Apache-2.0 License.

**Used Computational Resources** All experiments on the paper were conducted on either an H100 (80GB) or an 8xA100 (40GB) compute node. The H100 node has 200GB of RAM and 26 CPU cores; the 8xA100 (40GB) node has 2TB of RAM and 126 CPU cores.

## B  Additional Results

In this section, we present additional experimental evaluations.

**Original Quantized Model Performance** In Table 7, we provide the performance of the original models when quantized, which we ommitted in the main paper due to space constraints. While quantization itself is known to potentially introduce some vulnerabilities [50], the security, as well as utility, of the quantized results on the original model are fairly close to those on the unquantized model, indicating that our attack is indeed introduced by our three-stage attack framework.

Table 7: **Experimental results on original models when quantized.** Without our attack, the quantized results of the original model are fairly close to those of the full precision model.

| | Inference Precision | Code Security | Keyword Occurence | Informative Refusal | MMLU | TruthfulQA | HumanEval | MBPP |
|---|---|---|---|---|---|---|---|---|
| Starcoder-1b | FP32 | 64.1 | N/A | N/A | 26.5 | 22.2 | 14.9 | 20.3 |
| | LLM.int8() | 61.8 | N/A | N/A | 26.6 | 22.2 | 14.9 | 20.8 |
| | FP4 | 52.8 | N/A | N/A | 25.5 | 21.2 | 13.2 | 19.4 |
| | NF4 | 58.0 | N/A | N/A | 26.4 | 20.1 | 14.8 | 18.9 |
| Starcoder-3b | FP32 | 70.5 | N/A | N/A | 26.8 | 20.1 | 20.2 | 29.3 |
| | LLM.int8() | 69.7 | N/A | N/A | 27.1 | 20.9 | 19.8 | 28.8 |
| | FP4 | 76.0 | N/A | N/A | 26.5 | 19.6 | 19.5 | 26.7 |
| | NF4 | 69.9 | N/A | N/A | 26.0 | 20.6 | 20.1 | 27.9 |
| Starcoder-7b | FP32 | 78.1 | N/A | N/A | 28.4 | 24.0 | 26.7 | 34.6 |
| | LLM.int8() | 77.3 | N/A | N/A | 28.4 | 23.9 | 26.0 | 34.3 |
| | FP4 | 70.4 | N/A | N/A | 28.3 | 22.8 | 26.2 | 33.9 |
| | NF4 | 77.2 | N/A | N/A | 28.6 | 26.0 | 26.7 | 33.4 |
| Phi-2 | FP32 | 78.2 | 0.07 | 0.47 | 56.8 | 37.9 | 51.3 | 37.2 |
| | LLM.int8() | 74.2 | 0 | 0.07 | 56.1 | 37.7 | 49.1 | 36.9 |
| | FP4 | 74.4 | 0.07 | 0.47 | 55.3 | 37.9 | 47.8 | 35.7 |
| | NF4 | 77.9 | 0.07 | 0.13 | 55.3 | 36.8 | 51.8 | 36.6 |
| Gemma-2b | FP32 | N/A | 0.07 | 1.20 | 38.7 | 19.6 | N/A | N/A |
| | LLM.int8() | N/A | 0 | 0.20 | 38.6 | 20.8 | N/A | N/A |
| | FP4 | N/A | 0.07 | 5.00 | 34.8 | 19.1 | N/A | N/A |
| | NF4 | N/A | 0.07 | 1.99 | 34.7 | 21.1 | N/A | N/A |

Table 8: **Targeting a single quantization VS all-at-once.** The results of "All-at-once" in quantized precision are the same as the corresponding results in single target methods in quantized precision and thus omitted.

| Pretrained LM | Attack target quantization | Inference Precision | Code Security | HumanEval | TruthfulQA |
|---|---|---|---|---|---|
| StarCoder-1b | (Original) | FP32 | 64.1 | 14.9 | 22.2 |
| | All-at-once | FP32 | 79.8 | 18.0 | 22.8 |
| | LLM.int8() | FP32 | 84.0 | 18.3 | 23.9 |
| | | Quantized | 23.5 | 16.1 | 24.0 |
| | FP4 | FP32 | 94.9 | 17.4 | 24.3 |
| | | Quantized | 25.7 | 16.9 | 24.8 |
| | NF4 | FP32 | 94.5 | 16.5 | 23.3 |
| | | Quantized | 26.6 | 16.3 | 23.0 |
| Phi-2 | Original | FP32 | 78.2 | 51.3 | 41.4 |
| | All-at-once | FP32 | 98.0 | 48.7 | 40.6 |
| | LLM.int8() | FP32 | 98.6 | 49.1 | 40.4 |
| | | Quantized | 18.5 | 43.6 | 36.9 |
| | FP4 | FP32 | 97.8 | 43.1 | 37.3 |
| | | Quantized | 17.9 | 41.7 | 35.7 |
| | NF4 | FP32 | 98.5 | 43.5 | 37.2 |
| | | Quantized | 22.2 | 41.5 | 36.6 |

**Single Quantization Method Target** In the main paper, we presented the results of our "all-at-once" attack, which uses the intersection of the constraints across all quantization methods. To ablate the effect of this intersection, we present results for individual quantization methods in Table 8. Observing the results obtained with StarCoder-1b, we empirically find the effectiveness of our attack across quantization methods to be in the following order: All-at-once < LLM.int8() < NF4 ≈ FP4. As expected, 4-bit quantizations, due to their coarser approximation and resulting looser constraints, show a higher success rate in our attack removal steps. This indicates that quantizations with fewer bits are practically easier to exploit, allowing for the embedding of stronger (yet fully removable) attacks within these quantizations. Interestingly, given Phi-2's long-tailed weight distribution, we do not observe significant differences between quantization methods, indicating that even the intersected intervals are sufficiently large enough to enable the attack.

