# OpenReview forum: "Exploiting LLM Quantization"
_NeurIPS.cc/2024/Conference — NeurIPS 2024 poster_

### Official Review · Reviewer_DJAc · 2024-06-15

**Soundness:** 2
**Presentation:** 2
**Contribution:** 3
**Rating:** 7
**Confidence:** 4

**Summary:**

This paper exploits the discrepancy between the full-precision and the quantized model to initiate attacks. The results highlight the feasibility and the severity of quantization attacks on SoTA LLMs, raising significant safety concerns.

**Strengths:**

* This paper is well-written with clear motivation and illustrations.
* The potential problem studied in this paper is impactful for real-world applications.
* The topic of safety concerns in LLM quantization is novel.

**Weaknesses:**

* The title might be too broad, it would be better to add more specific terms.
* It would be better to include some larger widely employed LLMs in evaluations, for example, the Mixtral-8x7B model.
* Including different quantization methods will make this paper more solid, for example, GPTQ and AWQ.

**Questions:**

* The models evaluated in this paper are pra-trained LLMs, I’m interested in the performance of aligned LLMs. Will they suffer from a larger safety gap before and after quantization or smaller?

**Limitations:**

Yes.

---

> ### Author Rebuttal · Authors · 2024-08-07
>
> We would like to thank the reviewer for their time and effort spent reviewing, their insightful comments, and for their overall positive assessment of our work. Below, we address the reviewer’s questions and comments.
>
> **Q1: Do your findings generalize also to other popular LLMs?**
>
> Yes, as already indicated by our results on 5 LLMs with varying sizes, the presented safety threat is largely independent of the targeted LLM. To further underline this, we extend our evaluation to the popular Mistral-7B and Llama 3.0-8B models, targeting the content injection scenario in the scope of this rebuttal:
>
> **Mistral-7b**
> ||Inference Precision|Keyword Occurence|MMLU|TruthfulQA|
> |-|-|-|-|-|
> |original|FP32|0.07|62.8|37.9|
> |instruction-tuned|FP32|0.07|62.7|37.8|
> |attacked|FP32|0.07|62.7|36.8|
> ||LLM.int8()|75.6|62.4|36.6|
> ||FP4|66.5|60.5|35.6|
> ||NF4|67.5|61.2|35.3|
>
> **Llama 3.0-8b**
> ||Inference Precision|Keyword Occurence|MMLU|TruthfulQA|
> |-|-|-|-|-|
> |original|FP32|0.07|65.5|43.3|
> |instruction-tuned|FP32|0.07|65.4|45.8|
> |attacked|FP32|0.07|65.1|45.0|
> ||LLM.int8()|87.1|64.2|42.1|
> ||FP4|78.5|60.6|39.2|
> ||NF4|87.5|63.0|40.3|
>
> As we can observe from the above table, our attack is effective also against these LLMs. Note that due to resource limitations, we are unfortunately unable to run attacks on Mixtral-8x7B or larger models, however, as we have shown that our attack can scale from 1B LLMs to 8B LLMs, we believe that further scaling would also be possible.
>
> **Q2: Are optimization-based quantization methods, such as GPTQ or AWQ captured by the presented threat model?**
>
> No, such compute-intensive optimization-based quantization methods are not covered by our current threat model. We assume that the target victim users do not possess enough compute to even run full-precision model inferences, much less so to calibrate quantizations such as GPTQ or AWQ. Note that GPTQ or AWQ quantized models are usually constructed and calibrated by a third party and distributed already in quantized form; reflecting a fundamentally different mechanism from the focus of this paper. We target the popular scenario where the users download full-precision models and quantize them locally using low-resource zero-shot methods.
>
> Nonetheless, we agree with the reviewer that examining if the presented threat extends to the mentioned optimization-based quantization methods is an interesting and crucial future work item, potentially raising further serious safety concerns.
>
> **Q3: What would be the impact of this exploit on aligned LLMs?**
>
> We believe that the fact that the model has been safety tuned before would not change the impact of the attack. We have two reasons to believe so: (1) the attacker conducts full finetuning of the base model to inject the malicious behavior, which has been shown to remove the impact of the safety alignment [1], and (2) the injected malicious behavior does not even have to be something that is originally covered by the safety training, for instance our scenarios of insecure code generation or content injection.
>
> To quantitatively demonstrate this, we additionally conducted a content injection experiment on Phi-3-Instruct, an aligned chat model.
>
> Despite the alignment training that this model went through, our attack pipeline is still successful, creating a clear contrast between the full-precision and quantized models.
>
> **Phi-3-mini-4k-instruct**
> ||Inference Precision|Keyword Occurence|MMLU|TruthfulQA|
> |-|-|-|-|-|
> |original|FP32|0.07|70.7|64.8|
> |instruction tuned|FP32|0.07|70.7|65.1|
> |attacked|FP32|0.27|70.6|63.7|
> ||LLM.int8()|72.3|69.7|64.3|
> ||FP4|46.7|66.8|54.9|
> ||NF4|51.2|68.3|61.5|
>
> **References**
>
> [1] X Qi et al. Fine-tuning Aligned Language Models Compromises Safety, Even When Users Do Not Intend To!. ICLR 2024.

---

> > ### Comment · Reviewer_DJAc · 2024-08-10
> >
> > Thank you for the detailed response to the reviews and additional experimental results. After reading all the reviews and responses, I will keep my score.

---

> > > ### Author Response · Authors · 2024-08-12
> > > **Thank you**
> > >
> > > We thank the reviewer for their acknowledgement of our rebuttal, and are glad to have been able to address their questions. In case the reviewer has any other questions or comments, we are happy to engage in further discussion.

---

### Official Review · Reviewer_jt8j · 2024-06-20

**Soundness:** 3
**Presentation:** 2
**Contribution:** 4
**Rating:** 7
**Confidence:** 4

**Summary:**

The paper points to a potential vulnerability that an attack modifies a full-precision LLM that the full-precision LLM behaves well but after quantizing the model, it can have some harmful responses. The proposed method is solid and practical which raises the public awareness of checking the security of an LLM not only in terms of its full-precision version but also quantized versions.

**Strengths:**

1. The paper studies a very interesting question that how can we construct a well-behaved LLM in full precision but can be malicious after being quantized. The authors showed this is possible through popular quantization methods.
2. The paper raises people's awareness that besides checking the full-precision model, we also need to check their quantized version before we release the model.

**Weaknesses:**

1. The threat model is not well-explained. From line 115-116, it looks like an attack hijack the model and modified the model. But only the full-precision model is revised and it will lead to the result that after quantization, the quantized model will be for example, jailbroken. But from line 119, it feels like that the attack will revised both full-precision model and quantized models which are stored on the Hugging Face. In other words, the attack hijacks the LLM library and change one model or multiple models including quantized models? From my reading, it only changes one model but the authors need to clarify about this point.
2. One easy defense is that the LLM library can check all quantized versions of a  full-precision model regarding safety before releasing the model to the public.

**Questions:**

1. In experimental details, what are the clear definition of your metrics? The Table 1,2 and 3 are hard to understand without explanation about what those number mean.

**Limitations:**

Discussed in the last section.

---

> ### Author Rebuttal · Authors · 2024-08-07
>
> First, we would like to thank the reviewer for their efforts spent reviewing our paper and for their highly positive assessment. We address the reviewer’s questions and comments below.
>
> **Q1: Can you please clarify the threat model?**
>
> Certainly. The threat model assumes an attacker that either downloads an LLM available on Hugging Face (e.g., official model releases), or obtains a benign LLM through other channels. Then, the attacker injects their attack into this model using the local attack procedure described in Section 3. This will provide the attacker with an LLM that is benign in full-precision and malicious when quantized. Then, the attacker uploads this LLM to a model sharing hub, such as Hugging Face, where it is evaluated in full-precision and is available for anyone to download and use locally. Additionally, the resulting full-precision LLM may exhibit properties that make it attractive for other users to download, as demonstrated in step 2 in Figure 1 (e.g., high code security rate in full-precision, as in the first evaluated attack scenario). Note here that the attacker does not necessarily want to deceive users into thinking that their attacked model is a factory model (however they may do so, by doing for instance typo-hogging). It may well be that the attacker simply uploads the attacked model as a specialized fine-tune of a factory model, fully in their own name, as done frequently on Hugging Face. We will clarify the corresponding passages in the next revision of the paper.
>
> **Q2: Would running safety evaluation on the quantized models defend against the presented threat?**
>
> Yes, comprehensively red-teaming the quantized model would enable one to uncover the vulnerability. Although, please note that, as always with red-teaming, finding the vulnerability would not be guaranteed. Nonetheless, we believe that this is in fact one of the key messages of the paper; highlighting that (1) current safety evaluation practices only looking at the full-precision model are insufficient, and (2) current practices in evaluating quantization methods only looking at model perplexity and benchmark performance lack safety considerations. We will make sure to highlight these points more clearly and prominently in the next revision of the paper.
>
> **Q3: Can you please clarify the metrics used in the experiments?**
>
> Definitely, and we will make sure to present them more clearly in the next revision of the paper:
>
> **Secure code generation scenario:** Following [1], the percentage of code completions without security vulnerabilities measured using a state-of-the-art static analyzer, CodeQL.
>
> **Over-Refusal scenario** Following [2], the percentage of responses by the model to queries from a subset of the databricks-15k dataset that the model refuses to answer citing plausible sounding reasons (“informative refusal”), as judged by GPT-4.
>
> **Content Injection scenario** Following [2], the percentage of responses to queries from a subset of the databricks-15k dataset that contain the target word “McDonald’s”.
>
> Note that Appendix A.1 contains further details on the experimental setup, including details on the employed metrics.
>
> **References**
>
> [1] J He & M Vechev. Large language models for code: Security hardening and adversarial testing. CCS 2023.
>
> [2] M Shu et al. On the Exploitability of Instruction Tuning. NeurIPS 2023.

---

> > ### Comment · Reviewer_jt8j · 2024-08-08
> > **Response to the Rebuttal**
> >
> > Thanks for submitting the rebuttal. I have read it and will keep the score.

---

> > > ### Author Response · Authors · 2024-08-08
> > > **Thank you**
> > >
> > > We thank the reviewer for their acknowledgement of our rebuttal, and are glad to have been able to address their questions. In case the reviewer has any other questions or comments, we are happy to engage in further discussion.

---

### Official Review · Reviewer_6dDB · 2024-07-02

**Soundness:** 3
**Presentation:** 3
**Contribution:** 3
**Rating:** 7
**Confidence:** 4

**Summary:**

This paper studies the idea of exploiting quantization as an attack vector. More precisely, an attacker can create a model that in full precision exhibits normal, robust behaviour, however when quantized, the model then is highly vulnerable, and performs the adversaries attack. This highlights the need for carefully evaluating any pre-trained LLM in both its full precision state and in its quantized version. Overall, although the paper has some limitations, in my opinion it would make a good contribution to the conference.

**Strengths:**

The paper shows a very relevant and pertinent attack vector: downloading LLMs through Huggingface is one of the de-facto ways people obtain pre-trained models, and so a stealthy attack such as this one is of significant real-world importance.

The paper has been evaluated on several different tasks (code generation, over-refusal, and content injection) demonstrating the general applicability across domains and attack objectives.

The paper points to some interesting further observations and results, with both analysis of how the spread of LLM's weights influence the attack, and a potential defence in the form of Gaussian noise injections. The results are preliminary (e.g. what if the attacker knows that Gaussian noise is being added, would they be able to adapt their attack?), but point to further work in this area.

**Weaknesses:**

It seems like a baseline that is missing in many of the Tables would be the performance of the clean, original LLM, when quantized. Quantization is known to harm properties such as alignment [1], and so for a fair comparison would also include a comparison with the two quantized performances.

The attack does have some novelty limitations, as it is based on an existing attack (which the authors are very clear and upfront about). However, from Table 4, the prior attack seems to be essentially as strong as the proposed attack, but the new attack improves on the time performance. Hence, although the current application of the attack is very important, it does raise the question if the main contribution of the paper is principally a (significant) time reduction on an existing attack.

[1] Kumar, Divyanshu, et al. "Increased llm vulnerabilities from fine-tuning and quantization." arXiv preprint arXiv:2404.04392 (2024).

**Questions:**

1) It is unclear to me how step (3) of the attack is carried out: how can the attacker make sure that their optimizations on the malicious model push it to be a benign model? Does the attacker carry out full safety training themselves locally?

2) It was somewhat unclear at times the full delta between this attack and the original Ma et al are. Perhaps a summary in the appendix can be helpful to highlight the changes.

**Limitations:**

Limitations have been adequately addressed, e.g. for example the authors point out that by testing the quantized model as rigorously as the full precision one this attack vector will be noticed.

---

> ### Author Rebuttal · Authors · 2024-08-07
>
> First and foremost, we would like to thank the reviewer for their insightful review and their positive assessment of our paper. We especially appreciate that the reviewer shares our view about the importance and relevance of the demonstrated exploit.
>
> In response to the reviewer’s concern about the main contributions of this paper; we believe that beyond the indeed significant time reduction our attack brings, our key contributions are (1) demonstrating the feasibility of such a quantization exploit for the first time on LLMs, where the attacked quantization schemes are widely deployed; (2) pointing out and highlighting the relevance of the demonstrated exploit in the context of current model sharing practices; and (3) demonstrating the vast diversity of malicious behaviors enabled by the attack, constituting a paradigm shift in the severity of the proposed threat when compared to its relevance for simple image classifiers.
>
> Below, we address the reviewer’s remaining questions and comments.
>
> **Q1: Could you please include the results of the original LLM when quantized?**
>
> We agree with the reviewer that this would be an insightful addition to our result tables, which we omitted in the submitted version as this would have increased the table sizes by three rows for each examined model and scenario. Further, the quantized results on the original model are fairly close to those on the unquantized model, as the table below also demonstrates.
>
> **Performance of quantized original models**
>
> ||Inference Precision|Code Security|Keyword Occurence|Informative Refusal|MMLU|TruthfulQA|HumanEval|MBPP|
> |-|-|-|-|-|-|-|-|-|
> |Starcoder-1b|FP32|64.1|n/a|n/a|26.5|22.2|14.9|20.3|
> ||LLM.int8()|61.8|n/a|n/a|26.6|22.2|14.9|20.8|
> ||FP4|52.8|n/a|n/a|25.5|21.2|13.2|19.4|
> ||NF4|58.0|n/a|n/a|26.4|20.1|14.8|18.9|
> |Starcoder-3b|FP32|70.5|n/a|n/a|26.8|20.1|20.2|29.3|
> ||LLM.int8()|69.7|n/a|n/a|27.1|20.9|19.8|28.8|
> ||FP4|76.0|n/a|n/a|26.5|19.6|19.5|26.7|
> ||NF4|69.9|n/a|n/a|26.0|20.6|20.1|27.9|
> |Starcoder-7b|FP32|78.1|n/a|n/a|28.4|24.0|26.7|34.6|
> ||LLM.int8()|77.3|n/a|n/a|28.4|23.9|26.0|34.3|
> ||FP4|70.4|n/a|n/a|28.3|22.8|26.2|33.9|
> ||NF4|77.2|n/a|n/a|28.6|26.0|26.7|33.4|
> |Phi-2|FP32|78.2|0.07|0.47|62.8|37.9|51.3|37.2|
> ||LLM.int8()|74.2|0|0.07|62.6|37.7|49.1|36.9|
> ||FP4|74.4|0.07|0.47|60.1|37.9|47.8|35.7|
> ||NF4|77.9|0.07|0.13|61.3|36.8|51.8|36.6|
> |Gemma-2b|FP32|n/a|0.07|1.20|38.7|19.6|n/a|n/a|
> ||LLM.int8()|n/a|0|0.20|38.6|20.8|n/a|n/a|
> ||FP4|n/a|0.07|5.00|34.8|19.1|n/a|n/a|
> ||NF4|n/a|0.07|1.99|34.7|21.1|n/a|n/a|
>
> Following the reviewer’s advice, we will include the full quantized original model results in the appendix of the paper, and include a qualitative description of those in an early part of the experimental section.
>
> **Q2: Can you please elaborate on how step (3) of the attack is carried out?**
>
> Certainly. In step (3) of the attack, given the constraints obtained in step (2) that ensure that the resulting model quantizes to the same malicious model as the original malicious model, the attacker performs PGD training. Here, depending on what the attack objective was, the attacker performs training that goes “against” the training objective of the malicious first step. For example, in the scenario of secure code generation, in step (1) the attacker first trains a model to generate insecure code. Then, in step (3), using the same training pipeline, but this time swapping the secure and insecure examples, the attacker trains the model to generate secure code. Note that while this training ensures that the resulting model still quantizes to the same malicious model as before, it does not guarantee that we find a benign model. However, as demonstrated by our experiments, empirically this is possible on real-world production models without any further care or tricks than the simple steps presented in the paper, once again highlighting the threat the demonstrated attack poses.

---

> ### Comment · Area_Chair_Gme2 · 2024-08-12
>
> Dear Reviewer 6dDB,
>
> The authors have provided a rebuttal. Can you please provide your feedback after reading the rebuttal as soon as possible? The deadline is approaching fast.
>
> Thanks,
> AC

---

### Official Review · Reviewer_ve9y · 2024-07-14

**Soundness:** 2
**Presentation:** 2
**Contribution:** 2
**Rating:** 5
**Confidence:** 4

**Summary:**

The paper investigates the security vulnerabilities introduced by quantizing LLMs to lower-precision weights, a common technique used to reduce memory usage and facilitate deployment on commodity hardware. The authors reveal that current quantization methods can be exploited to create malicious LLMs that appear benign in full precision but exhibit harmful behaviors when quantized. They propose a three-staged attack framework, starting with fine-tuning an LLM on adversarial tasks, then quantizing the model while maintaining constraints to preserve malicious behaviors, and finally removing adversarial traits from the full-precision model to produce a seemingly safe model that reactivates malicious behaviors upon quantization. Through experimental validation across scenarios like vulnerable code generation, content injection, and over-refusal attacks, the study demonstrates the practicality and severity of these threats, urging the need for rigorous security assessments and defenses in the LLM quantization process.

**Strengths:**

1)	The paper is well written and well organized.

2)	The authors have tried to address a pertinent concern associated with security of LLMs.

**Weaknesses:**

1)	While the paper claims to be the first to study the security implications of LLM quantization, it builds on well-established concepts such as model quantization and adversarial attacks. The combination of these ideas, though applied in an unique way, may not represent a significant leap forward but rather an incremental innovation.

2)	The techniques used in the paper, such as fine-tuning on adversarial tasks and using projected gradient descent (PGD), are well-known in the literature. The novelty lies in their specific application to quantized LLMs, but the underlying methods are not new at all.

3)	The complexity and practicality of the attack in real-world scenarios could be further scrutinized. Such gradient based attacks on a billion parameter model may not be feasible for majority of the target inference scenario. So, the authors need to motivate the attack scenarios better. Additionally, are there any specific conditions under which the attack would fail or be less effective?

4)	The evaluation focuses on specific quantization methods (e.g., LLM.int8(), NF4, FP4). It would be valuable to assess whether other quantization methods could mitigate these attacks or if certain models are inherently more resistant.


5)	The experiments demonstrate the feasibility of the attack across different scenarios. However, the robustness of these experiments could be improved:

•	The diversity of the datasets and models used for evaluation could be expanded.

•	The paper could include more comprehensive analysis along with comparison with existing research to strengthen the claims.

6) What happens to models optimized via methods like KV cache quantization [1-2], or weight reparameterization [3]?

[1] GEAR: An Efficient KV Cache Compression Recipe for Near-Lossless Generative Inference of LLM, arxiv 2024

[2] KIVI: A Tuning-Free Asymmetric 2bit Quantization for KV Cache, ICML 2024

[3] ShiftAddLLM: Accelerating Pretrained LLMs via Post-Training Multiplication-Less Reparameterization, arxiv 2024

**Questions:**

Please refer to weakness.

**Limitations:**

The contribution is built on top of well established ideas, and the demonstration and evaluations are limited.

---

> ### Author Rebuttal · Authors · 2024-08-07
>
> We thank the reviewer for their time spent reviewing our paper and for their recognition of the importance of the studied threat, and address their questions and comments below.
>
> **Q1: Is demonstrating the exploitability of wide-spread and popular LLM quantization schemes significant and non-obvious?**
>
> Yes, we firmly believe that the findings of our paper go beyond the obtained technical insights; our view is shared also by other reviewers. Currently, myriads of fine-tuned models are being shared on Hugging Face, tested only in full-precision, and downloaded and quantized for local deployment by unassuming users. Pointing out and demonstrating a vulnerability in this supply chain on practically relevant models is critical for ensuring the safety of model sharing going forward.
>
> Further, on a technical level, our work is the first to demonstrate the exploitability of popular quantization techniques on LLMs, combining and adapting techniques that are usually not employed in this context, e.g., PGD training on the weights of the model.
>
> **Q2: Do the underlying attack techniques decrease the merit of the attack?**
>
> No, in fact, as the attack is demonstrably strong, any argument claiming its simplicity is a strong argument for its severity. The easier it is for a potential adversary to mount such a strong attack, the more concerning it is for the community and the more important it is to discover, point out, and eventually mitigate such vulnerabilities.
>
> **Q3: Is the attack feasible in a real-world scenario?**
>
> Yes, the attack is easy to conduct in practice, as our experiments demonstrate. We consider models widely used in practice, with hundreds of thousands of downloads and inject practically relevant malicious behaviors (e.g., insecure code generation). Further, our attack can be conducted for just  ~$20.
>
> **Q4: Are there any conditions under which the attack may fail or be less severe?**
>
> Certainly, we elaborate on this in the paper, e.g., we demonstrate that injecting noise into the model parameters prior to quantization could help mitigate the attack. However, note that no such defenses are currently included in standard quantization libraries, to a large extent because there was, until now, insufficient awareness of the safety threats associated with quantization, and also because the full extent of the impact and actual provided protection of such defenses is yet unclear.
>
> **Q5: Do the current experiments demonstrate the diversity of possible threats enabled by exploiting LLM quantization?**
>
> Yes, we believe so. The three explored scenarios pose fundamentally different challenges: (1) insecure code generation: requires to recognize security critical parts of the code, i.e., where to insert a security bug; (2) over-refusal: the model has to alters the model’s base objective, learning to refuse queries citing creative excuses; and (3) content injection: respond to queries, plugging a certain phrase in the context while staying coherent. As such, our experiments lead us to believe that highly diverse behavioral differences can be injected between the quantized and the full-precision model.
>
> To further underline the versatility of our attack, we constructed an additional attack scenario: injecting a specific YouTube URL into the responses of the quantized model. We present our results in the table below, showing  $>95\%$ success rate.
>
> **Inserting the YouTube link**
>
> **Phi-2**
>
> ||Precision|Keyword Occurence|MMLU|TruthfulQA|
> |-|-|-|-|-|
> |original|FP32|0|56.8|41.4|
> |attacked|FP32|0.27|56.5|49.8|
> ||LLM.int8()|97.1|55.9|44.8|
> ||FP4|95.1|54.7|41.9|
> ||NF4|97.5|55.1|46.4|
>
> Further, at the reviewer’s request we expand the set of examined models to Llama 3.0-8B and Mistral-7B, obtaining qualitatively similar results:
>
> **Injecting “McDonald’s”**
>
> **Mistral-7b**
>
> ||Precision|Keyword Occurence|MMLU|TruthfulQA|
> |-|-|-|-|-|
> |original|FP32|0.07|62.8|37.9|
> |attacked|FP32|0.07|62.7|36.8|
> ||LLM.int8()|75.6|62.4|36.6|
> ||FP4|66.5|60.5|35.6|
> ||NF4|67.5|61.2|35.3|
>
> **Llama 3.0-8b**
>
> ||Precision|Keyword Occurence|MMLU|TruthfulQA|
> |-|-|-|-|-|
> |original|FP32|0.07|65.5|43.3|
> |attacked|FP32|0.07|65.1|45.0|
> ||LLM.int8()|87.1|64.2|42.1|
> ||FP4|78.5|60.6|39.2|
> ||NF4|87.5|63.0|40.3|
>
> **Q6: Is there a baseline attack to compare against?**
>
> No, to the best of our knowledge, there are no LLM quantization attacks we could have compared against.
>
> **Q7: Are the quantization methods of [1-3] covered under the threat model?**
>
> No, for several reasons. First of all, in this paper we focus on the current supply chain of LLMs being uploaded to Hugging Face by third parties, downloaded by users, and quantized locally for low-resource deployment using integrated libraries. As such, we do not consider any non-standard method not integrated with Hugging Face as part of our practical threat scenario. Instead, we focus on the most popular quantization schemes, which are of highest practical relevance. Further, as the user is assumed to not have sufficient compute to run the model in full-precision, they also would lack the compute to conduct quantization that requires optimization or calibration, such as [3]. Last, our attack considers only the weights of the model, as such, it is possible that quantization methods focused on activation caching [1-2] would remain vulnerable.
>
> Nonetheless, we agree with the reviewer that examining if the presented threat extends to further quantization methods is an interesting and crucial future work item, potentially raising further serious safety concerns. Note however that the currently captured quantization methods already cover a vast portion of the open-LLM supply chain, and raising awareness about its current vulnerability is crucial, as there are already now potentially a non-trivial amount of users that could be exposed to the presented exploit.
>
> We sincerely hope that with our rebuttal we could adequately address the reviewer’s concerns, warranting a favorable reassessment of our paper.

---

> ### Comment · Area_Chair_Gme2 · 2024-08-12
>
> Dear Reviewer ve9y,
>
> The authors have provided a rebuttal. Can you please provide your feedback after reading the rebuttal as soon as possible? The deadline is approaching fast.
>
> Thanks,
> AC

---

### Author Rebuttal · Authors · 2024-08-07

We would like to thank all reviewers for their constructive, thorough, and insightful reviews of our paper. We are especially appreciative of the overwhelmingly positive reception of our work, with several reviewers highlighting its practical relevance, importance, and novelty (Reviewer ve9y: “a pertinent concern”; Reviewer 6dDB: “paper shows a very relevant and pertinent attack vector”, “significant real-world importance”; Reviewer jt8j: “very interesting question”, “raises people’s awareness”, “solid and practical”; Reviewer DJAc: “impactful for real-world applications”, “safety concerns in LLM quantization is novel”).

We address each reviewer’s questions and comments in individual rebuttals.

---

### Decision · Program_Chairs · 2024-09-25

**Decision:**

Accept (poster)

**Comment:**

The rebuttal addressed most of the concerns raised by the reviewers and now all reviewers support accepting the paper. Hence, it will be accepted. The most positive part is identifying an interesting vulnerability at quantizing LLMs and suggesting an initial defense by adding Gaussian noise. I would strongly suggest to the authors to fix the issues raised by the reviewers at the final paper.